# Effective Natural Killer Cell Degranulation Is an Essential Key in COVID-19 Evolution

**DOI:** 10.3390/ijms23126577

**Published:** 2022-06-13

**Authors:** Sara Garcinuño, Francisco Javier Gil-Etayo, Esther Mancebo, Marta López-Nevado, Antonio Lalueza, Raquel Díaz-Simón, Daniel Enrique Pleguezuelo, Manuel Serrano, Oscar Cabrera-Marante, Luis M. Allende, Estela Paz-Artal, Antonio Serrano

**Affiliations:** 1Instituto de Investigación Sanitaria Hospital 12 de Octubre (imas12), 28041 Madrid, Spain; garcinunosara@gmail.com (S.G.); fgile@salud.madrid.org (F.J.G.-E.); esther.mancebo@salud.madrid.org (E.M.); martalopezne@gmail.com (M.L.-N.); antonio.lalueza@salud.madrid.org (A.L.); dpleguezuelo@salud.madrid.org (D.E.P.); msblanco@salud.madrid.org (M.S.); oscar.cabrera@salud.madrid.org (O.C.-M.); luis.allende@salud.madrid.org (L.M.A.); estela.paz@salud.madrid.org (E.P.-A.); 2Department of Immunology, Hospital Universitario 12 de Octubre, 28041 Madrid, Spain; 3Department of Internal Medicine, Hospital Universitario 12 de Octubre, 28041 Madrid, Spain; rdiazs@salud.madrid.org; 4Department of Immunology, Ophthalmology and Otorhinolaryngology, Facultad de Medicina, Universidad Complutense de Madrid, 28040 Madrid, Spain; 5Biomedical Research Centre Network for Epidemiology and Public Health (CIBERESP), 28029 Madrid, Spain

**Keywords:** COVID-19, natural killer, innate immunity, degranulation activity, granzymes

## Abstract

NK degranulation plays an important role in the cytotoxic activity of innate immunity in the clearance of intracellular infections and is an important factor in the outcome of the disease. This work has studied NK degranulation and innate immunological profiles and functionalities in COVID-19 patients and its association with the severity of the disease. A prospective observational study with 99 COVID-19 patients was conducted. Patients were grouped according to hospital requirements and severity. Innate immune cell subpopulations and functionalities were analyzed. The profile and functionality of innate immune cells differ between healthy controls and severe patients; CD56dim NK cells increased and MAIT cells and NK degranulation rates decreased in the COVID-19 subjects. Higher degranulation rates were observed in the non-severe patients and in the healthy controls compared to the severe patients. Benign forms of the disease had a higher granzymeA/granzymeB ratio than complex forms. In a multivariate analysis, the degranulation capacity resulted in a protective factor against severe forms of the disease (OR: 0.86), whereas the permanent expression of NKG2D in NKT cells was an independent risk factor (OR: 3.81; AUC: 0.84). In conclusion, a prompt and efficient degranulation functionality in the early stages of infection could be used as a tool to identify patients who will have a better evolution.

## 1. Introduction

COVID-19 is an infectious disease caused by severe acute respiratory syndrome coronavirus 2 (SARS-CoV-2) with a wide spectrum of clinical profiles [1]. Severe forms of COVID-19 have been directly associated with an immune system dysregulation [2] that includes a profound abnormal activation of both CD4+ and CD8+ T cell compartments [3,4,5] with a massive release of pro-inflammatory cytokines (cytokine storm). This massive release triggers an uncontrolled immune response, damage to the lung tissue, multiorgan failure [6,7,8,9,10], and impaired type I interferon (IFN) responses [11]. The reasons an abnormal immune response occurs are not well understood. One possible hypothesis is that immune dysregulation is associated with the presence of comorbidities and with the predominant immune response in the initial moments of the disease [12]. It has been observed that rapid Th1 responses are associated with lower probabilities for the development of hyperinflammatory complications, which are frequently observed in patients with an initial Th2 and/or Th17 immune response [13,14].

The study of immune dysregulation in COVID-19 patients has focused on adaptive immunity. Lymphopenia, activated and senescent effector phenotypes, and an impaired Th response are determinants for the outcome of the disease [3,4,13]. Innate immunity plays a crucial role in the primary non-antigen-specific immune responses against infection [15]. Lymphoid and myeloid cells are found in the innate immune system that includes dendritic cells, natural killer cells (NK), natural killer T cells (NKT), mucosal-associated invariant T cells (MAIT), γδTCR T cells, neutrophils, eosinophils, basophils, and monocytes, among others [16,17,18].

Innate immunity is also affected in COVID-19 patients. It has been found that a poor evolution of the disease is associated with decreased levels of proteins of the innate immune system such as the apolipoprotein H, a molecule related to the clearance of apoptotic cells and viral particles from the blood [19]. In addition, a correlation has been reported between the blockage of human interferon-α by auto-antibodies and severe forms of the disease [20]. However, little is known about the cellular and functional abnormalities.

In the early phase of viral infection, NK cells are the first lymphocytes to respond to pathogens, contributing to the activation of innate immunity through the release of cytokines. This rapid response occurs before the development of an adaptive immune response [21,22]. The NK cells are able to recognize viral-infected cells by pattern-recognition membrane receptors that trigger inhibitory and activation signals. The activation procedure is caused by the stimulation of activation receptors with stress-induced molecules or viral proteins expressed on the cell surface. Another activation pathway is through a reduced expression of class I human leukocyte antigen (HLA) on the surface of the neighboring cells. The presence of class I HLA is an inhibitory signal for NK cells [23,24,25]. 

One of the best hallmarks of NK activation is the beginning of its effector activity, which can be evidenced by its degranulation. This is a phenomenon in which the secretion of granzymes and perforins induces cytotoxicity through the activation of the caspases’ cascade of infected cells that are forced to enter apoptosis [26,27]. 

Innate immunity dysfunction has been strongly associated with viral and tumor susceptibility. This fact has been well-characterized by inborn errors of immunity such as MAGT1 deficiency and hemophagocytic lymphohistiocytosis (HLH). MAGT1 is a magnesium transporter involved in protein glycosylation, one of the major post-translational modifications. The hypoglycosilation caused by mutations in the MAGT1 gene abolishes the expression of NKG2D, an activating receptor necessary for NK and T-cell activation. Its deficiency has been associated with uncontrolled Epstein–Barr infection and lymphoma development [28,29]. On the other hand, impaired NK degranulation against viral infection is associated with hemophagocytic lymphohistiocytosis (HLH), an uncommon severe systemic inflammatory syndrome that causes a strong activation of the immune system with hypercytokinemia, multiorgan failure, and poor prognosis [30]. In an attempt to counteract the ineffective NK activity, other cells in the immune system (T cells and macrophages) undergo a sustained hyperactivation process damaging tissues and blood cells [31]. Secondary HLH is the consequence of the over-reactive immune response against a trigger, such as a viral infection, which could be accompanied by diminished degranulation activity [32]. Some authors have reported that secondary HLH is the result of a transitory reduction in the cytotoxic activity (associated with genetic polymorphisms) [33].

Initially, COVID-19 was connected with secondary HLH [34]. Laboratory similarities such as levels of ferritin, levels of sIL-2R, and the activation of T cells and macrophages seemed to show a dysfunction in innate immunity [35,36,37]. As a final event, the hyperactivation of T cells and macrophages could produce huge amounts of pro-inflammatory cytokines involved in the cytokine storm [38]. 

Impaired degranulation in other infections such as human immunodeficiency virus (HIV), recurrent herpes simplex virus (HSV), and Toxoplasma gondii has been well-characterized [26,39,40]. However, the implications for innate immunity in the physiopathology of COVID-19 have not been widely studied. This work has aimed to characterize the innate immune-cell profile of COVID-19 patients in the acute phase of the disease and to study their NK cell activity and its association with clinical progression.

## 2. Results

### 2.1. Patient Characteristics, Lymphocyte Subpopulations, and Inflammatory Parameters

The cohort of COVID-19 patients had a median age of 49 years (IQR: 36.2–59) with a homogenous gender distribution (male: 54%, *p* = 0.365). No significant differences regarding age were observed when the COVID-19 cohort was compared to the healthy controls (median age 49 years vs. 49 years, respectively; *p* = 0.557). Non-hospitalized patients were significantly younger than hospitalized ones (median age 43 years vs. 53 years, respectively; *p* = 0.001) without significant differences in sex distribution (male 54% vs. 46, *p* = 0.260). 

The median percentage of CD3+ T cells was significantly lower in COVID-19 patients than in healthy controls: 62.7% (51.8–68.8) vs. 69.9% (62.2–72.2), *p* = 0.004 (Table 1). Similar results were obtained when the non-hospitalized COVID-19 patients were compared to the severe ones: 64.2% (59.7–73.2) vs. 55.1 % (48–64), *p* = 0.009 (Appendix A). No significant differences were observed when CD4+ and CD8+ subpopulations were analyzed.

The analysis of the major blood cell subpopulations showed that non-hospitalized patients had a higher total number of lymphocytes and a higher median percentage of CD3+ T cells compared to hospitalized patients: 1300 lymphocytes/μL (1000–1600) vs. 900 (600–1425), *p* = 0.002 and 64. 2% (59.7–73.2) vs. 58.1% (48–67.4), *p* = 0.004 for the CD3+ T cells percentage (Table 2). The study of the inflammatory parameters (Table 2) also showed differences between non-hospitalized and hospitalized COVID-19 patients. Patients who were hospitalized had higher median levels of CRP: 7.44 mg/dL (2.1–11.3) vs. 1.18 (0.4–2.8), *p* < 0.001 and LDH: 359 U/L (314–428) vs. 261 (213–31), *p* < 0.001. The DD levels were significantly higher in hospitalized patients compared to non-hospitalized patients: 674 ng/dL (241–1429) vs. 516 (387–645), *p* = 0.024. Nevertheless, the analysis of DD was biased since the number of patients in whom the DD levels were studied was low as DDs were only tested in those patients with a severe clinical process.

COVID-19 patients were grouped according to the development of ARDS and then compared. Non-severe patients showed lower lymphopenia compared to severe patients, including the total number of lymphocytes, with 1200 (800–1600) vs. 950 (600–1300 cells/uL), *p* = 0.067 (Appendix A) and the median percentage of CD3+ cells, with 63.4% (53.8–70.5) vs. 55.1% (48–64), *p* = 0.013 (Appendix A).

### 2.2. Innate Immune Profile in COVID-19

The in-depth analysis of the innate immune cell compartments revealed that COVID-19 patients showed a marked increased tendency in the median percentage of NK cells (evaluated as CD3-CD56+ cells) compared to healthy controls: 14.3% (8.5–19.6) vs. 9.2% (7.2–14.9), *p* = 0.051 (Table 1). 

However, the percentage of NK cells reached significance when severe COVID-19 patients were compared to healthy controls: 18% (8.5–25.5) vs. 9.2% (7.2–14.9), *p* = 0.021 (Figure 1A). When the NK cells were divided according to CD56 expression in NKbright (CD3-CD56++, immunoregulatory principally through cytokine production) and NKdim (CD3-CD56+, cytotoxic activity), we observed that COVID-19 patients presented lower median percentages of CD56bright NK and higher percentages of CD56dim NK cells compared to healthy controls: 0.4% (0.2–6) vs. 0.55% (0.4–0.8), *p* = 0.016 for CD56bright NK cells and 13.6% (8.2–19) vs. 8.7% (6.6–14.4), *p* = 0.039 for CD56dim NK cells (Table 1). These CD56bright and CD56dim NK cells were also reduced and increased, respectively, when non-hospitalized and severe patients were compared to healthy controls: 0.4% (0.3–0.6) vs. 0.55% (0.4–0.8), *p* = 0.046 for CD56bright NK cells (Figure 1B) and 17.25% (8.6–25.1) vs. 8.7% (6.6–14.4), *p* = 0.023 for CD56dim NK cells (Figure 1C).

The study of mucosal-associated invariant T (MAIT) cells (CD3 + Vα7.2 + CD161+), T cells with a semi-invariant αβ TCR that display innate effector-like qualities, showed a significant reduction in the percentage in the COVID-19 patients compared to the healthy population including the total MAIT, with a median of 0.9% (0.4–2.3) vs. 2.85% (1.6–4.15), *p* < 0.001 (Table 1); and those gated from CD8, with 1.8% (1.7–4.3) vs. 4.4% (2.2–11), *p* = 0.001 (Table 1), but not for those expressing CD4. This same scenario was observed when healthy controls were compared to non-hospitalized patients, with 2.85% (1.6–4.15) vs. 1.4% (0.5–3.3), *p* = 0.037; and to the severe ones, with 2.85% (1.6–4.15) vs. 0.6% (0.5–2.3) *p* = 0.009 for total MAIT cells (Figure 1D). The same trend was observed when CD8+ MAIT cells from healthy controls were compared to non-hospitalized patients: 4.6% (2.2–11) vs. 2.2% (0.8–7.2) *p* = 0.021 and the severe counterparts: 4.6% (2.2–11) vs. 1.5% (0.8–4.3) *p* = 0.008 for CD8+ MAIT cells (Figure 1E).

In addition, the COVID-19 cohort was studied alone. At that time, the innate immune profile of non-hospitalized patients in comparison to their hospitalized counterparts only showed a discrete increase in the median percentage of CD4+ MAIT cells: 0.4% (0.3–1.2) vs. 0.3% (0.1–0.7; *p* = 0.036; data not shown). This same tendency was observed in all the MAIT cells: 1.4% (0.5–3.3) vs. 0.6% (0.3–1.9), *p* = 0.052.

No significant comparisons between the NKT, MAIT, and γδT cells are shown in Appendix A.

### 2.3. NKG2D and CD107a Expression in NK and NKT Cells

The expression (medium fluorescence intensity, MFI) of NKG2D was evaluated in NK and NKT (CD3 + CD56+) cells. This is an activating receptor that can trigger cytotoxicity and its density in the cell membrane of NK and NKT cells could represent their activation status. COVID-19 patients showed a diminished expression of NKG2D in NK cells compared to healthy controls: 32,256 (27,210–39,459) vs. 39,129 (34,876–50,420), *p* < 0.001 (Table 1). This phenomenon was repeated when healthy controls were compared to non-hospitalized and severe COVID-19 patients, with 39,192 (34,876–50,420) vs. 33,330 (28,672–37,952), *p* = 0.006 for non-hospitalized patients; and 32,545 (27,410–39,884), *p* = 0.007 for severe patients (Figure 2A). Similar results were obtained when the same analysis was performed to study the expression of NKG2D in NKT cells with values of 99,577 (81,873–107,068) in healthy controls vs. 62,247 (45,737–82,792) in COVID-19 patients (Table 1), *p* < 0.001; 60611 (42,408–76,618) in non-hospitalized patients (*p* < 0.001); and 70,436 (56,316–96,138) in severe COVID-19 patients, *p* = 0.009 (Figure 2B). No significant comparisons of NKG2D are shown in Appendix A.

CD107a, which is exposed during the degranulation process, is a molecule present in the inner membrane of exocytosis granules. To evaluate the degranulation activity of NK cells, the expression of CD107a was measured by the MFI fold change after a co-culture with stimulatory cells (lacking HLA-I). No significant statistical significance was found when the COVID-19 cohort was analyzed in comparison to the healthy controls (Table 1). However, when COVID-19 was divided according to disease severity, it was observed that severe COVID-19 patients presented an impaired NK cell degranulation activity compared to healthy controls: 7.6 (5.5–10) vs. 11 (9.8–17.4), *p* = 0.005 (Figure 2C). The degranulation activity of NK cells showed that severe patients presented impaired granule exocytosis compared to non-severe patients: 7.6 (5.5–10) vs. 10.2 (6.9–15.39), *p* = 0.009 (Figure 2D). Similar results were observed when asymptomatic and severe patients were compared (Appendix A).

An example of CD107a and NKG2D MFI in NK cells in the healthy controls, non-hospitalized, and severe COVID-19 patients is shown in Figure 2E,F respectively.

### 2.4. Granzyme A and B Studies in COVID-19 Patients

The evaluation of the cytotoxic activity of the NK cells was also evaluated by measuring the plasma levels of granzyme A and B. No significant differences were observed when the levels of both granzymes in the COVID-19 patients were compared to the healthy controls (Appendix A) or when the patients were divided according to disease severity (Appendix A). However, when the granzyme secretion was evaluated as a ratio between plasmatic granzymes A and B (granzyme ratio), it was found that non-hospitalized COVID-19 patients presented a higher granzyme ratio than severe patients: 114.7 (53.9–271.2) vs. 37.5 (27.9–62.7), *p* = 0.013 (Figure 3A). Similarly, the granzyme ratio observed in non-severe COVID-19 patients was significantly higher than that observed in those who developed severe forms of the disease: 102 (50.4–301.2) vs. 37.5 (27.9–62.7), *p* = 0.001 (Figure 3B).

In order to study the association between granzyme secretion and degranulation activity, we divided the COVID-19 patients according to low and high degranulation levels and observed that there was a correlation between those patients with impaired degranulation activity and a significant reduction in the secretion of granzyme A: 3900.4 pg/mL (2768.5–10,040.2) vs. 13887.2 (7,986.5–27,507.6), *p* = 0.018 (Appendix A). This same finding was observed when the granzyme ratio was analyzed: 52.14 (37.8–85.7) vs. 138.4 (83.2–241.7), *p* = 0.009 (Figure 3C).

### 2.5. Multivariable Analysis

In order to determine the relevance of the type of immune response in COVID-19 severity, five different multivariate analyses were performed that included those innate variables that had shown a *p* value < 0.1 in a previous univariate analysis. 

The first multivariate analysis conducted identified the total count of lymphocytes (OR: 0.34, 95% CI: 0.12–0.95, *p* = 0.041) in the early immune response to SARS-CoV-2 infection and a higher expression of NKG2D in NKT cells (OR: 2.02, 95% CI: 1.1–3.9, *p* = 0.033) as a protective and risk factor, respectively, for hospitalization requirements with an area under the curve ROC of 0.779 (95% CI AUC: 0.683–0.56, Table 3A).

A second multivariate analysis was conducted according to the presence of severe forms of the disease (Table 3B). The expression of NKG2D in NKT cells (OR: 2.22, 95% CI: 1 1.12–4.4, *p* = 0.022) was identified as a risk factor for the development of severe forms of the infection. Nevertheless, the degranulation activity behaved as a discrete protective factor for the development of ARDS (OR: 0.87, 95% CI: 0.78–0.98, *p* = 0.021) with an area under the curve ROC of 0.752 (95% CI AUC: 0.655–0.834).

A third multivariate analysis was performed to study the differences between non-hospitalized patients and those patients who required hospitalization with mild to moderate symptoms (Table 3C). The total number of lymphocytes (OR: 0.26, 95% CI: 0.08–0.81, *p* = 0.017) was significant and was an independent protective factor with an area under the curve ROC of 0.803 (95% CI AUC: 0.695–0.886).

A fourth multivariate analysis studied the differences between non-hospitalized patients and severe forms of the disease (Table 3D). NKG2D expression in NKT cells (OR: 3.51, 95% CI: 1.44–8.53, *p* = 0.005) was an independent risk factor for the development of severe forms of the disease. However, the degranulation activity (OR: 0.86, 95% CI: 0.75–0.99, *p* = 0.047) resulted in a significant and independent protective factor, all together with an area under the curve ROC of 0.840 (95% CI AUC: 0.729–0.918).

A final multivariate analysis studied which parameters were associated with asymptomatic and severe forms of the disease (Table 3E). The total number of lymphocytes (OR: 0.14, 95% CI: 0.02–0.87, *p* = 0.032) and the degranulation activity (OR: 0.84, 95% CI: 0.72–0.98, *p* = 0.033) in the early stage of infection were found to be significant and independent protective factors, with an area under the curve ROC of 0.808 (95% CI AUC: 0.627–0.927).

## 3. Discussion

In this study, we have been able to demonstrate that the intensity of the early innate immune response is related to the severity of the disease. Interestingly, the analysis of NKG2D in NKT cells showed that a higher expression correlated with greater disease severity. Furthermore, the balance of the degranulation activity, measured as the MFI fold change, behaved as an independent protective factor for the development of severe forms of the disease. This scenario suggests that the intensity of the initial degranulation activity in COVID-19 could be of paramount importance for the control of the disease, as has been described in other infectious diseases such as HIV, Toxoplasma gondii, and frequently recurring HSV [26,39,40]. Additionally, as far as we know, this study represents the first time that the granzymes ratio has been analyzed. This parameter has demonstrated that patients with non-severe forms of COVID-19 showed higher granzyme ratios than those who developed severe forms.

As previously reported, COVID-19 patients have presented abnormalities in the inflammatory markers when they have been evaluated according to disease progression. Our results are consistent with those facts. We found that levels of LDH, CRP, and DD were higher in those patients with a worse evolution, validating the importance of inflammation in the pathology of the disease [41]. 

COVID-19 patients who did not require hospital admission presented a normal total count of lymphocytes, whereas both hospitalized and severe patients showed profound lymphopenia. These results are in line with other published results that demonstrate that these lymphocytes could be used as a predictive marker for disease severity [42,43]. In this work, similar results were found when we compared non-hospitalized patients to their hospitalized counterparts. 

Changes in the lymphoid compartment may be involved in the immunopathophysiology and evolution of COVID-19. Mucosal-associated invariant T (MAIT) cells are abundant in organs such as the liver or gut accounting for more than 50% and 12%, respectively, of the total lymphocytes in these locations. MAIT cells constitute a part of the innate immune system, which mediates anti-bacterial and anti-viral responses [17,44]. It is known that MAIT cells play a major role in eradicating intracellular bacterial infections such as *Klebsiella* or mycobacteria. Although their activation implies antigenic recognition through a minor complex of histocompatibility, MAIT cells could be stimulated by different cytokines such as IL-12, IL-15, IL-18, or IFNα/β. In different infections, MAIT cells suffer an expansion and activation to clear the microorganism. However, it has been observed that those cells suffer a reduction compared to healthy donors in chronic viral infections such as dengue or HVC [45,46]. 

In SARS-CoV-2 infection, we have observed that the MAIT compartment suffered a deep reduction in COVID-19 patients compared to healthy controls. This phenomenon was more evident when healthy controls were compared to COVID-19 patients divided according to disease severity, similar to what Parrot et al. reported [47]. The significant decrease in MAIT cells could be caused by the migration of peripheral cells to the inflamed tissues. Likewise, the reduction in the proportion of CD3 cells in severe COVID-19 patients is not only related to a decrease in the proportion of CD8 cells but also to a mild decrease in the proportion of CD4 cells and a significant decrease in MAIT cells. Depending on the expression of CD56 (N-CAM) in the NK cell membrane, CD56bright or CD56dim, NK cells present different functions. The former are cells with immunoregulatory potential, some of them being efficient cytokine producers, whereas the latter present huge cytotoxic activity [48]. Osman et al. published that the NK compartment showed substantial differences between COVID-19 patients and healthy controls. Similar to our data, they observed that COVID-19 patients presented higher percentages of CD56dim and lower percentages of CD56bright NK cells [49]. 

NK and NKT cells express a huge variety of activating receptors, NKG2D being one of the most important [50,51]. The regulatory balance to mount an effective anti-viral response is reached through different stimuli [52]. NKG2D plays an important role in viral and tumor clearance via cell degranulation [53]. The study of the expression of NKG2D revealed that COVID-19 patients showed an impaired activation status and cytotoxic capacity compared to healthy controls. This phenomenon was clearer when healthy controls were compared to COVID-19 patients divided according to disease severity, as occurs in other viral infections [54]. On the other hand, the study of the expression of this activation marker in NKT cells in all COVID-19 cohorts showed that the expression of NKG2D in NKT cells may be an independent risk factor for the development of severe forms of the disease. A possible explanation for this might be that NKT cells could be persistently activated with an inefficient function and protection against the infection. This status would maintain continuous production of cytokines, as a compensatory mechanism, involved in the physiopathology of the disease [13,55]. 

The degranulation process against viral infection is essential for the correct clearance of an infection via the secretion of perforin and granzyme among other enzymes [56]. Alterations of this procedure are associated with viral susceptibility. The study of the degranulation activity in COVID-19 patients demonstrated that severe patients presented a dysfunctionality not only comparable with their non-severe counterparts but also with asymptomatic patients. These results are consistent with other published results where reduced apoptosis of K562 cells in severe and critical patients compared to mild ones was observed [57,58]. However, those works just compare COVID-19 patients with mild/moderate to severe forms of the disease, whereas this study analyzed the most distant clinical groups of patients, from patients without symptoms to those who were critical. This fact demonstrates the importance of innate functionality in the initial phase of the pathology. In addition, we have reported that COVID-19 patients who did not require hospital admission showed similar degranulation activity compared to healthy controls, suggesting that correct degranulation activity is mandatory for the resolution of the condition. Therefore, efficient innate degranulation activity in the initial moments of the infection can prevent the development of severe forms of the disease. Mazzoni et al. reported that the presence of low levels of granzymes and perforins resulted in alterations in the degranulation activity in severe patients with higher levels of serum IL-6. However, after the administration of biological treatments such as tocilizumab, the concentration of those cytotoxic enzymes was normalized [59,60]. 

In addition to their function as effector cells of innate immunity, NK cells play a fundamental role in the modulation of adaptive immune responses [61]. It has been described that NK cells instruct dendritic cells to promote Th1 immunity during intracellular infections [62]. In the same way, NK cell depletion leads to reduced Th1 responses [63]. Our group recently described the importance of a strong Th1 response for a positive outcome of COVID-19 [14]. This fact together with the findings described in this work suggests that adequate NK activity, both as an effector and modulator towards the Th1 response, would be essential to establish a coordinated cytotoxic response of innate and adaptive immunity, which allows for the prompt clearance of SARS-CoV2 infection. Likewise, the cytokines produced by Th1 cells would provide positive feedback to NK cells’ functionality, reinforcing the activity of innate and adaptive cellular immunity.

This impaired process of degranulation activity resulted in diminished secretions of granzymes and perforins and the subsequent apoptosis of the infected cells. Based on the granzyme ratio between granzyme A and granzyme B, we have observed that COVID-19 patients with non-severe forms of the disease presented a higher capacity for granzyme A secretion and a lower capacity for granzyme B secretion, contrary to their severe counterparts. However, no differences were found when the extracellular levels of each one were evaluated separately. These results are in line with those reported by other groups where a higher expression of granzyme B correlates with severity of disease [60,64]. The use of a granzyme ratio to analyze the extracellular levels of granzymes allows for the normalization of differential activity. As is well-known, both granzyme A and granzyme B molecules play an important role in independent and dependent caspase cell-mediated apoptosis, respectively [65]. NK and CD8 + T cells are the major producers of these two proteases, although granzyme A is produced in greater amounts than granzyme B [66,67]. Nevertheless, the extracellular activity of granzymes is completely different from the intracellular one [68]. Although the gold standard in the study of granzymes is the intracellular flow cytometry, in this work, soluble granzymes in serum were evaluated with the ELISA methodology, measuring the levels of the extracellular granzymes. This method offers a panoramic and systemic view of the situation in the whole body. However, the intracellular analysis only studies the circulating lymphocytes and not the in situ scenario in the tissues and organs.

The higher concentrations of granzyme A in the plasma of mild COVID-19 patients could reflect the parallel activity of the innate and adaptive cytotoxic pathways. Furthermore, it has been published that higher extracellular levels of granzyme A correlate with peaks in the circulation levels of IFN-γ [69,70]. This reflects the induction of a correct and efficient Th1 immunity against viral infection. Other authors have postulated that granzyme A contributes to the clearance of the virus through a pro-inflammatory environment, which inhibits intracellular viral replication [71,72]. Another important function of extracellular granzyme A is its matrix-remodeling activity. This fact contributes to the migration of cytotoxic T lymphocytes to the infected location [66,68]. All of this leads us to believe that granzyme A is an important mediator in viral infection to achieve its correct clearance. On the contrary, extracellular levels of granzyme B have been involved in several chronic pathologies and have been associated with tissue and organ injury [73]. There are several studies in which high levels of extracellular granzyme B have been associated with cardiac injury, including acute myocardial infarction, aortic aneurysm, or transplant vasculopathy [74,75,76]. As granzyme B is elevated in chronic and persistent inflammatory diseases, its role could be used to define their severity and physiopathology. 

In light of these results, NK-directed therapy increasing their functionality would be a potential treatment for SARS-CoV-2 infection as it occurs in other infectious diseases. The use of genetic engineering such as CAR-NK cells directed to SARS-CoV-2 peptides could be a promising therapeutic tool [77].

The main limitation of this work is the size of the cohort as some groups are small when patients are divided according to severity. Another limitation is that this is a single-center study. The results should be validated in subsequent multicenter studies that include a larger number of patients.

## 4. Materials and Methods

### 4.1. Study Design

A prospective observational study that recruited patients in the early stages of COVID-19 was conducted in a tertiary university hospital in Spain. 

Peripheral innate cells including MAIT, NKT, γδT cells, and NK cells and their activation and degranulation activity were examined at the time of diagnosis. Hospitalized patients were followed-up until discharge or death. Clinical information of non-hospitalized patients was obtained in the Emergency Department.

### 4.2. Patients

A prospective observational study was conducted by recruiting a cohort of 101 COVID-19 patients in a random manner in the Emergency Department of the Hospital Universitario 12 de Octubre (Madrid, Spain) from 17 May to 7 September 2021. Inclusion criteria were (1) adult patients (>18 years) with high suspicion of SARS-CoV-2 infection, (2) confirmed COVID-19 diagnosis by RT-PCR in the early acute phase of the disease, and (3) follow-up until discharge or death. Two patients were excluded as they were lost to the follow-up. Finally, 99 COVID-19 patients were enrolled in the study. During follow-up, 96 patients recovered and 3 died. Asymptomatic patients were identified in the emergency department as close contacts of other relatives with symptoms. They were followed up to assess the possible appearance of symptoms.

A control group made up of 24 anonymous blood donors was created to compare innate populations and functionality. Control patients were negative for SARS-CoV-2 after antigen or PCR tests at blood donation time.

### 4.3. Patients Classification

Four groups of COVID-19 patients were created in accordance with the most critical event during the disease: groups 1 and 2 comprised non-hospitalized COVID-19 patients (*n* = 38), 16 (group 1) of whom were asymptomatic and 22 (group 2) of whom who were symptomatic; and groups 3 and 4 comprised hospitalized COVID-19 patients (61), 37 (group 3) of whom were hospitalized without complications and 24 (group 4) severe patients who developed acute respiratory distress syndrome (ARDS) as the main complication. Non-severe COVID-19 patients were those who did not develop ARDS (groups 1–3) (Figure 4).

### 4.4. Study Definitions

A COVID-19 case was defined as a patient suspected of having a SARS-CoV-2 infection after returning a positive result for a SARS-CoV-2 reverse transcription-polymerase chain reaction (RT-PCR) assay performed on a nasal swab sample. 

Ventilatory failure was defined as a SaO_2_/FiO_2_ < 300 (blood oxygen pressure/fractional inspired oxygen).

Poor outcome was defined as patients who fulfilled at least one of the following criteria: (a) ventilatory failure, (b) admission to the intensive care unit (ICU), or (c) death during admission by any cause.

### 4.5. Data Collection

The patient data, including clinical, laboratory, and demographic data, were obtained from electronic medical records. Laboratory parameters included D-dimers (DD), lactate dehydrogenase (LDH), C reactive protein (CRP), and the number of lymphocytes.

### 4.6. Samples

Plasma and EDTA-treated blood samples were collected and processed in the first 24 h after admission to the Emergency Department. Admission occurred at 6 days (median) from the onset of symptoms in symptomatic COVID-19 patients.

### 4.7. Innate Cells Subsets

EDTA-treated whole blood was incubated using the corresponding monoclonal antibodies: anti-CD3-PC5.5 and anti-CD4-APC-A750 (all from Beckman Coulter, Miami, FL, USA); anti-TCRγδ-PE, anti-NKG2D-PCy7 and anti-CD8-APC (all from BDBiosciences, Franklin Lakes, NJ, USA); and anti-CD56-FITC and anti-CD161-FITC (all from BioLegend, San Diego, CA, USA). Innate subsets were analyzed by flow cytometry using a Dx-Flex Cytometer and Kaluza Software (Beckman Coulter, Miami, FL, USA).

NK and NKT cells were considered CD3-CD56+ and CD3 + CD56+, respectively, gated from lymphocytes. MAIT cells were considered Vα7.2+ and CD161+ gated from CD3 + αβTCR+ T cells. MAIT cells expressing CD4 and CD8 were gated from CD3 + CD4+ or CD8+ lymphocytes, respectively, expressing both Vα7.2 and CD161.

### 4.8. NK Degranulation Assay

Peripheral blood mononuclear cells were isolated by the Ficoll gradient method. Mononuclear cells were stimulated overnight (37 °C, 5% CO_2_) with IL-2 (100 U/mL, Roche, Basel, Switzerland). These cells were then co-cultured with K562 in 0:1 and 1:1 ratios for 4 h (37 °C, 5% CO_2_) in a complete RPMI 1640 medium. Monensin (Golgi inhibitor) and anti-CD107a-PE (BDBiosciences, Franklin Lakes, NJ, USA) were added after the first hour. Cells were harvested after the culture were incubated with the following antibodies: anti-CD3-PCy5.5 (Beckman Coulter, Miami, FL, USA) and anti-CD56-FITC (BioLegend, San Diego, CA, USA) [78]. Degranulation assay was analyzed by flow cytometry using a Dx-Flex Cytometer and Kaluza Software (Beckman Coulter, Miami, FL, USA). 

Degranulation activity was measured as the CD107a MFI fold change (MFI stimulated/MFI non-stimulated) in CD56 + CD3-gated cells.

### 4.9. Granzyme Evaluation

ELISA-based immunoassays were used to determine the extracellular plasma concentrations of granzyme A and granzyme B (Human Granzyme A/B, ELISABASIC kit, Mabtech, Sweden). The experimental procedure was performed following the manufacturer’s recommendations.

A ratio of granzyme A to granzyme B: GA (pg/mL/GB (pg/mL) was obtained to define the relative presence of these two molecules. This parameter was created in our laboratory in order to normalize the extracellular secretion of each granzyme. The evaluation of extracellular granzymes was performed in 90 of the 99 COVID-19 patients due to the absence of plasma samples.

### 4.10. Statistical Analysis

Discrete variables were represented as a percentage and an absolute frequency. Chi-square test or Fisher’s exact test were used to study the association between qualitative variables. The odds ratio expressed the relative measure of an effect.

The median accompanied by the interquartile range (IQR) in brackets were used to represent continuous variables. The Mann–Whitney U test was used for comparisons between the two groups. Multivariate analyses were performed using a logistic regression model with variables that presented a *p* value < 0.1 in a previous univariate analysis. The variables having a high level of dispersion were classified by ranges.

We considered a *p*-value under 0.05 as a significant result of the analysis. Data were analyzed with MedCalc for Windows version 19.8 (MedCalc Software, Ostend, Belgium).

## 5. Conclusions

COVID-19 patients who evolve from an anecdotal infection to mild disease are those who establish an intense and effective degranulation process in the early stages of the disease. On the contrary, those patients with a complex evolution showed early defective degranulation activity, which translated into a diminished granzyme ratio. Hence, the evaluation of the NK functionality and the measurement of extracellular granzymes ratios could be used as a prognostic tool for the evolution of the disease by identifying those patients who will have a better evolution of the disease and contributing to its control.

The search for new therapies and vaccines that could boost and stimulate cytotoxic NK activity from the initial diagnosis could favor the management of severe COVID-19.

## Figures and Tables

**Figure 1 ijms-23-06577-f001:**
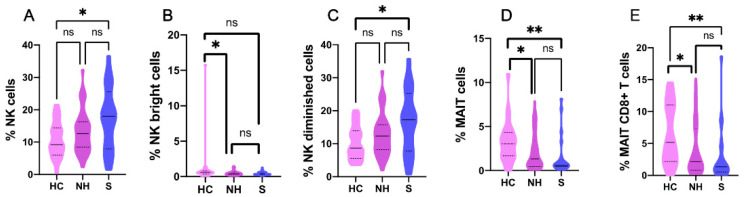
Innate immunological profile distribution in healthy controls (HC, *n* = 21), non-hospitalized (NH, *n* = 38), and severe COVID-19 patients (S. *n* = 24). (**A**) NK cells; (**B**) NK bright cells; (**C**) NK dim cells; (**D**) CD3+ MAIT cells; (**E**) CD8+ MAIT cells. HC, healthy controls; NH, Non-Hospitalized; S, Severe; ns, no significant; *, *p* < 0.05; **, *p* < 0.01.

**Figure 2 ijms-23-06577-f002:**
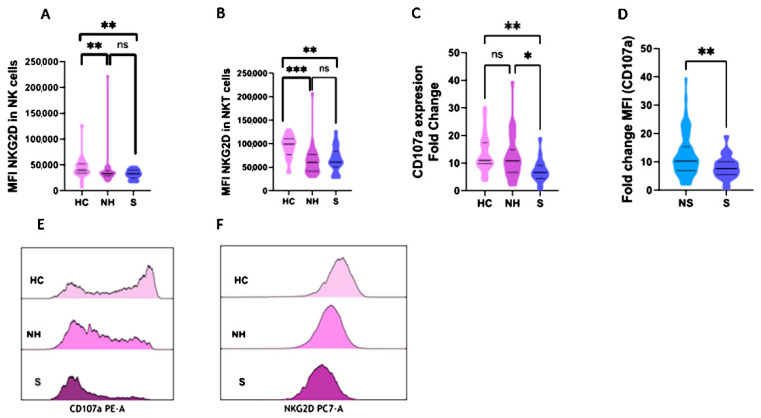
NK functionality in healthy controls (HC, *n* = 21), non-hospitalized (NH, *n* = 38), non-severe (NS, *n* = 75), and severe COVID-19 patients (S, *n* = 24). (**A**) MFI of NKG2D in NK cells; (**B**) MFI of NKG2D in NKT cells; (**C**,**D**) Fold Change (MFI CD107a); (**E**) CD107a MFI in NK cells, distribution for HC, NH, and S patients; (**F**) NKG2D MFI in NK cells, distribution for HC, NH, and S patients. HC, healthy controls; NH, Non-Hospitalized; NS, Non-severe; S, Severe; *, *p* < 0.05; **, *p* < 0.01; ***, *p* < 0.001.

**Figure 3 ijms-23-06577-f003:**
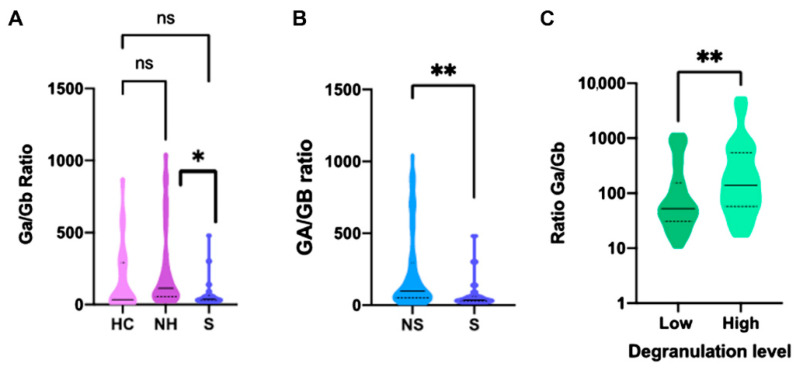
Granzyme ratio secretion according to disease severity. (**A**) Ga/Gb Ratio in HC, NH, and S COVID-19 patients; (**B**) Ga/Gb Ratio in NS and S COVID-19 patients; (**C**) Ga/Gb Ratio according to degranulation activity. HC, Healthy controls; NH, Non-hospitalized; NS, Non-severe; S, Severe. Low, CD107a expression Fold Change <9%; High, CD107a expression Fold Change >9%; *, *p* < 0.05; **, *p* < 0.01.

**Figure 4 ijms-23-06577-f004:**
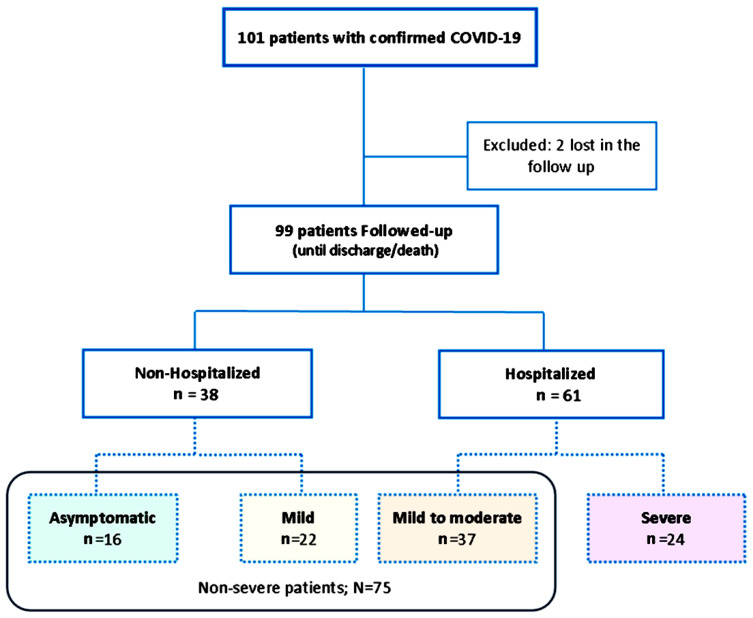
Algorithm of patient classifications according to disease severity.

**Table 1 ijms-23-06577-t001:** Innate immunological profile in healthy controls compared to COVID-19 patients.

	Healthy Controls; *n* = 24		COVID-19 Population; *n* = 99		
Variables	Median	IQR	Median	IQR	*p*-Value
% CD3+	69.9	62.2–73.2	62.7	51.8–68.8	0.004
% CD4+ in CD3+	65	59.2–68	60.1	54–69.2	0.262
% CD8+ in CD3+	31.8	23.3–38.2	36	28.7–42.9	0.222
% NK	9.2	7.2–14.9	14.3	8.5–19.6	0.051
% NK CD56 ^brigth^	0.55	0.35–0.8	0.4	0.2–0.6	0.016
% NK CD56 ^dim^	8.7	6.6–14.4	13.6	8.2–19	0.039
% NKT	4.6	1.7–9.8	4.9	2.9–7.5	0.904
% MAIT	2.85	1.6–4.2	0.9	0.4–2.3	0.001
% MAIT in CD4+ T cells	0.4	0.3–0.9	0.4	0.2–0.8	0.419
% MAIT in CD8+ T cells	4.6	2.2–11	1.8	1.7–4.3	0.001
MFI NKG2D in NK cells	39,192	34,876–50,420	32,256	27,210–39,459	<0.001
MFI NKG2D in NKT cells	99,577	81,873–107,068	62,247	45,737–82,792	<0.001
% TCR gd	4	2.7–10.1	3.8	2.2–6	0.275
CD107a Fold Change in NK cells	11	9.8–17.4	10	6.4–13.7	0.13

NK: Natural killer cells; NKT: Natural killer T cells; MAIT: mucosal-associated invariant T cells; MFI: Medium fluorescence intensity.

**Table 2 ijms-23-06577-t002:** Population characteristics in non-hospitalized and hospitalized COVID-19 patients.

	Non-Hospitalized; *n* = 38	Hospitalized COVID-19; *n* = 61	
Variables	Median	Median	*p*-Value
Male (%)	18 (47%)	36 (59%)	0.26
Female (%)	20 (53%)	25 (41%)	
Age (Years)	43 (32–50)	53 (38–62)	0.001
Lymphocytes (cells/uL)	1300 (1000–1600)	900 (600–1425)	0.002
Neutrophils (×10^3^ cells/uL)	3.8 (2.5–5.3)	5 (3.7–7.2)	0.019
CD3+ T lymphocytes (%)	64.2 (59.7–73.2)	58.1 (48–67.4)	0.004
CRP (mg/dL)	1.18 (0.4–2.8)	7.44 (2.1–11.3)	<0.001
LDH (U/L)	261 (213–31)	359 (314–428)	<0.001
DD (ng/dL; *n* = 64)	516 (387–645) (*n* = 17)	674(241–1429) (*n* = 47)	0.024

CRP: C-Reactive Protein; LDH: Lactate dehydrogenase; DD: D-Dimer.

**Table 3 ijms-23-06577-t003:** Innate risk factors associated with disease severity.

Variables	Univariant			Multivariant		
	OR	OR 95% IC	*p*-Value	OR	OR 95% IC	*p*-Value
(A) NH vs. H						
Lymphocytes	0.28	0.11–0.69	0.005	0.34	0.12–0.95	0.041
%CD3+	0.42	0.21–0.86	0.017	0.53	0.23–1.2	0.133
MFI NKG2D in NKT	2	1.1–3.6	0.022	2.02	1.1–3.9	0.033
Area Under the ROC Curve					0.779	0.683–0.856
(B) NS vs. S						
%CD3+	0.48	0.24–0.96	0.036	0.53	0.25–1.1	0.083
MFI NKG2D in NKT	2	1.01–3.8	0.036	2.22	1.12–4.4	0.022
CD107a expression in NK (MFI Fold change)	0.88	0.8–0.97	0.015	0.87	0.78–0.98	0.021
Area Under the ROC Curve					0.752	0.655–0.834
(C) NH vs. M						
Lymphocytes	0.26	0.1–0.71	0.008	0.26	0.08–0.81	0.017
MFI NKG2D in NKT	1.87	0.95–3.65	0.065	1.93	0.9–4.13	0.089
Area Under the ROC Curve					0.803	0.695–0.886
(D) NH vs. S						
MFI NKG2D in NKT	2.77	1.28–5.99	0.009	3.51	1.44–8.53	0.005
CD107a expression in NK (MFI Fold change)	0.89	0.8–0.99	0.031	0.86	0.75–0.99	0.047
Area Under the ROC Curve					0.84	0.729–0.918
(E) A vs. S						
Lymphocytes	0.23	0.05–1.02	0.053	0.14	0.02–0.87	0.032
CD107a expression in NK (MFI Fold change)	1.42	1.1–1.81	0.005	0.84	0.72–0.98	0.033
Area Under the ROC Curve					0.808	0.627–0.927

MFI: Medium fluorescence intensity; NKT: Natural killer T cells; NK: Natural killer cells.

## Data Availability

The raw data supporting the conclusions of this article will be made available by the authors, without undue reservation.

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
