# Peer review of "Effective Natural Killer Cell Degranulation Is an Essential Key in COVID-19 Evolution"

_ijms, 2022, doi:10.3390/ijms23126577_

Round 1

Reviewer 1 Report

The study by Garcinuno and colleagues aimed to characterize the relative abundance and functional capacity of NK and T cells in COVID-19 patients. The authors present clinical data as well as flow cytometry and ELISA data from blood samples of the patients. Overall, the study is well done, and provides some valuable information about immune responses to SARS-CoV-2 infection. However there are some issues that should be addressed:

1. In Table 2 data for D-dimer is presented for a subset of patients in the sample, but the number per group is not noted.

2. Age is significantly different between the non-hospitalized and hospitalized groups. Given the well known association of COVID-19 with age and immune function declining with age, some effort should be made to control for age in the analyses.

3. Throughout the paper the authors present cell percentages or MFI data from flow cytometry experiments. It would add some value to include representative flow plots of some of the data so that the reader can evaluate the data directly. This would be helpful especially for situations involving a subtle peak shift, such as the NKG2D data, or where ratios and fold changes are reported, such as CD107a and granzymes.

4. The granzyme ratio that the authors propose is interesting but requires further detail. Since this measurement is not well established in the literature, the authors should provide some validation of it using some historical cohort data. The granzyme ration data in Figure 3A and 3B appear to be highly skewed, so at a minimum the authors should analyze for outliers and remove them if necessary. Also, the comparison of granzyme ratio across degranulation level presented in Figure 3C should ideally be done as a simple linear regression, not a 2-sample comparison.

5. The multivariate analyses presented give some good insights into the potential clinical application of the data. However, presenting multiple binomial logistic regression models can be confusing to interpret. It may be more effective to us a single multinomial logistic regression and coding the dependent variable as it is detailed in Figure 4, i.e. asymptomatic, mild, moderate, severe.

6. Deaths and recoveries are not reported for the cohort.

Author Response

We appreciate the reviewer's comments; their observations will allow us to improve the aspects that hinder the correct comprehension of the manuscript. Below we will answer the questions point by point. In the new version of the manuscript have been included all the suggestions and clarifications requested by the reviewers, modifications are marked with “track changes” function. The manuscript has been reviewed by an English-born person.

  1. In Table 2 data for D-dimer is presented for a subset of patients in the sample, but the number per group is not noted.

Authors’ reply: We appreciate the request for clarification the read of the manuscript. The reviewer 1 is true about the affirmation. We just added the number of all patients tested for D-Dimer. We have modified the Table 2 in manuscript adding the number of non-hospitalized (N=17) and hospitalized (N=47) tested for that parameter separately.

  1. Age is significantly different between the non-hospitalized and hospitalized groups. Given the well-known association of COVID-19 with age and immune function declining with age, some effort should be made to control for age in the analyses.

Authors’ replay: We really appreciate the observation of the reviewer 1. In the previous manuscript we did not include the comparison of age between COVID-19 patients and healthy controls, since no significant differences were observed. The median age of each group was the same, 49 years. However, this has been included in the manuscript.

“No significant differences regarding age was observed when the COVID-19 cohort was compared to healthy controls (median age 49 years vs. 49 years, respectively; p=0.557).”

  1. Throughout the paper the authors present cell percentages or MFI data from flow cytometry experiments. It would add some value to include representative flow plots of some of the data so that the reader can evaluate the data directly. This would be helpful especially for situations involving a subtle peak shift, such as the NKG2D data, or where ratios and fold changes are reported, such as CD107a and granzymes.

Authors’ reply: We appreciate the indication of reviewer 1 according to the comprehension of MFI analysis. As the reviewer suggested, we have added in figure 2 two overlays of the flow cytometry MFI analysis (as an example). These images could help for the evaluation of the differences in MFI between healthy controls, hon-hospitalized and severe patients. The example of granzymes is not possible to be performed since we have done by ELISA, measuring extracellular levels of granzymes.

“An example of CD107a and NKG2D MFI in NK cells changes in healthy controls, non-hospitalized and severe COVID-19 patients is shown in figure 2E and 2F respectively.”

  1. The granzyme ratio that the authors propose is interesting but requires further detail. Since this measurement is not well established in the literature, the authors should provide some validation of it using some historical cohort data. The granzyme ration data in Figure 3A and 3B appear to be highly skewed, so at a minimum the authors should analyze for outliers and remove them if necessary. Also, the comparison of granzyme ratio across degranulation level presented in Figure 3C should ideally be done as a simple linear regression, not a 2-sample comparison.

Authors’ reply: We appreciate the request for clarification the read of the manuscript. In this moment we do not have the reagent needed for that purpose. But we consider that a validation and confirmation of these results are recommendable so, we will confirm it in further analysis. Regarding the skewed graphs in granzymes ratio, we have re-evaluated the comparison and outliers have been removed without compromising the significance. Then a total of 90 patients have been tested.

The evaluation of the cytotoxic activity of the NK cells was also evaluated measuring plasma levels of granzyme A and B. No significant differences were observed when the levels of both granzymes in COVID-19 patients were compared to healthy controls (table S2) or when the patients were divided according to disease severity (figure S5A-B). However, when the granzyme secretion was evaluated as a ratio between plasmatic granzymes A and B (granzyme ratio), it was found that non-hospitalized COVID-19 patients presented a higher granzyme ratio than severe patients: 114.7 (53.9-271.2) vs. 37.5 (27.9-62.7) p=0.013 (Figure 3A). Similarly, the granzyme ratio observed in non-severe COVID-19 patients was significantly higher than that observed in those who developed severe forms of the disease: 102 (50.4-301.2) vs. 37.5 (27.9-62.7) p=0.001 (Figure 3B).

About the simple linear regression in figure 3C, granzyme ratio did not follow a normal distribution (both Shapiro-Wilk and Kolmorogov-Simirnov test). Therefore, when performing the simple linear regression, the results are not significant. In this case just 2-sample comparison is possible using no-parametric Man-Whitney test.

  1. The multivariate analyses presented give some good insights into the potential clinical application of the data. However, presenting multiple binomial logistic regression models can be confusing to interpret. It may be more effective to us a single multinomial logistic regression and coding the dependent variable as it is detailed in Figure 4, i.e. asymptomatic, mild, moderate, severe.

Authors’ reply: We are grateful for the suggestion, but once we performed the analysis separating patients in asymptomatic, mild, moderate, severe each group is not large enough to achieve statistical significance. In addition, we considered that making this bigger groups gathering more general forms of the disease could have a bigger clinical repercussion in the management of COVID-19 patients, since we study extremes of the pathology.

In fact, we have performed the analysis as the reviewer said without statistical significance.

  1. Deaths and recoveries are not reported for the cohort.

Authors’ reply: We are very grateful for the carefully reading of the manuscript. In the first time we count the number of deceased and recovered patients and we observed a very tiny sample size of death COVID-19 patients, 3 patients. That is the reason why we did not include the statistical analysis in the manuscript. Nevertheless in patients’ classification we have included the number of death patients.

“Finally, 99 COVID-19 patients were enrolled in the study. During follow-up, 96 patients recovered and 3 died. Asymptomatic patients were identified in the emergency department as close contacts of other relatives with symptoms. They were followed up to assess the possible appearance of symptoms.”

Reviewer 2 Report

Sara Garcinuño and colleagues present a quality and well-written experimental manuscript describing effective natural killer cell degranulation is an essential key in COVID-19 evolution.

Authors aimed to characterize the innate immune cell-profile of COVID-19 patients in the acute phase of the disease and to study their NK cell activity and its association with the clinical progression. In their study authors conducted a prospective observational study with 99 COVID-19 patients that were grouped according to hospital requirements and severity. Innate immunity cells subpopulations and functionality were analyzed. 

Authors observed that the profile and functionality of innate immunity cells differs between healthy controls and severe patients, CD56dim NK cells being increased and MAIT cells and NK-degranulation rates decreased in COVID-19 subjects. High degranulation rates were observed in non-severe patients and in healthy controls compared to severe patients. Benign forms of the disease had a higher GranzymeA/GranzymeB ratio than complex forms. In a multivariate analysis, degranulation capacity resulted a protective factor against severe forms of the disease, whereas permanent expression of NKG2D in NKT cells was an independent risk factor.

Authors demonstrated that the intensity of the early innate immune response is related with the severity of the disease. Interestingly, the analysis of NKG2D in NKT cells showed that a higher expression correlated with greater disease severity. Furthermore, the balance of the degranulation activity, measured as the MFI fold change, behaved as an independent protective factor for the development of severe forms of the disease. This scenario suggests that the intensity of the initial degranulation activity in the COVID-19 could be of paramount importance for the control of the disease, as has been described in other infectious diseases like HIV, toxoplasma gondii and frequently recurrent HSV.

Overall, authors established that an early and efficient degranulation functionality at early stages of the infection could be used as a tool to identify patients who will have a better evolution. They conclude that the evaluation of the NK functionality and the measurement of extracellular granzymes-ratio could be used a prognostic tool for the evolution of the disease by identifying which patients will have a better evolution of the disease and contributing to its control.

Finally, authors suggest that the search for new therapies and vaccines which could boost and stimulate cytotoxic NK activity at the initial diagnosis could favor the management of severe COVID-19.

Overall, the manuscript is highly valuable for the scientific community and should be accepted for publication after the corrections are made.

==============================

Other comments:

1) Please check for typos throughout the manuscript.

2) Authors are kindly encouraged to cite the following article that describes the use of immune cell therapies against COVID-19 and other infections.
DOI: 10.3390/biomedicines9010059

Author Response

We appreciate the reviewer's comments; their observations will allow us to improve the aspects that hinder the correct comprehension of the manuscript. Below we will answer the questions point by point. In the new version of the manuscript have been included all the suggestions and clarifications requested by the reviewers, modifications are marked with “track changes” function. The manuscript has been reviewed by an English-born person.

1) Please check for typos throughout the manuscript.

Authors’ reply: We really appreciate the carefully reading of the manuscript, we have revised the manuscript looking for typos and we have corrected them. All the changes are marked with track changes.

2) Authors are kindly encouraged to cite the following article that describes the use of immune cell therapies against COVID-19 and other infections.
DOI: 10.3390/biomedicines9010059

Authors’ reply: We are grateful for the suggestion. We have cited the article and we agree with the reviewer 2 that by adding the citation we improve the value of the results presented in this work.

“At the light of these results NK directed therapy increasing their functionality would be a potential treatment in SARS-CoV-2 infections as it occurs in other infectious diseases. The usage of genetic engineering such as CAR-NK cells directed to SARS-CoV-2 peptides could be a promising therapeutic tool for the prognosis of the disease [78]”
